# Improving Conversational Recommendation Systems via Bias Analysis and Language-Model-Enhanced Data Augmentation

**Xi Wang**[†] **Hossein A. Rahmani**[†] **Jiqun Liu**[‡] **Emine Yilmaz**[†]

[†]University College London, London, UK
[‡]The University of Oklahoma, OK, USA

{xi-wang,hossein.rahmani.22,emine.yilmaz}@ucl.ac.uk
jiqunliu@ou.edu

## Abstract

Conversational Recommendation System (CRS) is a rapidly growing research area that has gained significant attention alongside advancements in language modelling techniques. However, the current state of conversational recommendation faces numerous challenges due to its relative novelty and limited existing contributions. In this study, we delve into benchmark datasets for developing CRS models and address potential biases arising from the feedback loop inherent in multi-turn interactions, including selection bias and multiple popularity bias variants. Drawing inspiration from the success of generative data via using language models and data augmentation techniques, we present two novel strategies, 'Once-Aug' and 'PopNudge', to enhance model performance while mitigating biases. Through extensive experiments on ReDial and TG-ReDial benchmark datasets, we show a consistent improvement of CRS techniques with our data augmentation approaches and offer additional insights on addressing multiple newly formulated biases.

## 1 Introduction

Conversational Recommendation System (CRS) is a growing research topic and application area, along with the recent advance in Natural Language Processing (NLP) and Conversational techniques. In contrast to traditional recommendation approaches, which provide item suggestions in a non-interactive manner, conversational recommendation involves multi-turn and mix-initiative interactions between users and the system (Jannach and Chen, 2022). Hence, a growing body of research studies (Zhang et al., 2018b; Chen et al., 2019; Li et al., 2022) has introduced diverse conversational recommendation models, which address the natural language understanding from user utterances, user intent estimation, user preference estimation and generation of appropriate responses that encapsulate the recommended items.

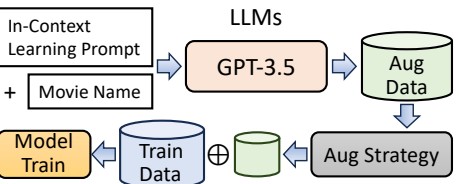

Figure 1: Data Augmentation Pipeline

Hence, in the development of conversational recommenders, we identify two primary streams of research directions: optimising natural responses and improving recommendation results. While recent advancements in language models have proven effective in generating natural utterances (Wang et al., 2022), enhancing recommendation results based on user preferences inferred from their utterances remains a challenge. These include skewed data interactions with limited coverage and missing evaluation based on the multi-turn interaction nature of CRS. In addition, due to the nascent nature of conversational recommendation techniques, there has been limited investigation into uncovering the potential biases within the feedback loop of their development. Left unchecked, the biases could lead to unfair information presentation, ineffective explorations, and poor decisions.

To address this research gap, we first characterise various potential biases in conversational recommendation systems, including selection bias in data and popularity-oriented biases based on the multi-turn interaction nature of CRS. Through a detailed analysis of CRS models, we identify these biases and their impact. Subsequently, by leveraging the advancements in Large Language Models (LLMs) and in-context learning approaches, we introduce novel data augmentation techniques, namely Once-Aug and PopNudge, to enhance the recommendation results and mitigate the effects of biases. Figure 1 provides an overview of the data augmentation pipeline that we applied in this study, which illustrates the generation of synthetic data followed

by the application of our novel augmentation strategies. These strategies enrich the training data, leading to improved performance of CRS models. To validate the effectiveness of our data augmentation strategies, we conduct extensive experiments using popular CRS baselines, such as KGSF (Zhou et al., 2020a) and KBRD (Chen et al., 2019), on two benchmark datasets: ReDial (Li et al., 2018) and TGReDial (Zhou et al., 2020b). The experimental results consistently demonstrate the performance improvement achieved by our data augmentation strategies, particularly PopNudge. Specifically, we evaluate the improvement based on recommendation accuracy, selection bias as well as various popularity-oriented biases and effects.

The main contributions of this paper are threefold: (1) a comprehensive investigation into potential biases and effects affecting the performance of CRS models, (2) novel data augmentation strategies for improving CRS models from multiple perspectives, and (3) extensive experiments demonstrating the effectiveness of the proposed data augmentation strategies and providing additional insights with discussion on novel biases and effects.

## 2 Related Work

This section offers an overview of the biases from data collection and the development of recommendation systems to the conversational recommendation system. In addition, we further discuss the existing bias mitigation strategies that leverage generative data or so-called data imputation.

### 2.1 Biases in Recommendation System

A recommendation *model* is designed to suggest items of high interest to a given *user* based on their preferences learned from historical interactions and associated information in the collected *data*. However, when examining the feedback loop involving the interactions between the user, data, and the model, it becomes apparent that recommender systems can suffer from various biases (Chen et al., 2022; Mehrabi et al., 2021; Wang et al., 2023; Salutari et al., 2023; Liu, 2023). For example, biased interactions between users and items, known as selection bias (Hernández-Lobato et al., 2014; Steck, 2013)) or the biased representation of items (i.e., exposure bias (Liu et al., 2020)) can result in skewed data distribution that makes it challenging to accurately capture user interests and ensure fair item representations. In particular, the commonly investigated popularity bias (Zhao et al., 2022; Naghiaei et al., 2022a), which arises from the concern about the growing applications of recommendation systems and the Matthew effect (cf. Wang et al., 2018) that reinforce interactions with popular items. Consequently, several strategies have been proposed to mitigate or alleviate bias effects in developing recommendation techniques (Wang et al., 2021; Liu et al., 2021; Naghiaei et al., 2022b). For example, Wang et al. (2021) addresses selection bias by validating the learned model on a small set of unbiased item ratings to enforce unbiased rating predictions.

In the context of Conversational Recommendation Systems (CRSs), which are the focus of this study, there is a growing concern about biases within the feedback loop, but limited research has been conducted to explore and mitigate these biases (Gao et al., 2021). Similar to the concerns regarding popularity bias in conventional recommendation systems, Lin et al. (2022a) and Fu et al. (2021) have investigated and contributed to mitigating popularity bias in CRSs. In addition to popularity bias, Shen et al. (2023) have explored the impact of unintended biases, such as racial and gender biases, on CRS recommendations. However, existing studies have relied on semi-synthetic conversations generated using user-posted reviews and developed using the system ask-user response method (Zhang et al., 2018b), rather than using real recommendation-oriented conversations to examine and evaluate bias effects. As a result, research on the effect of potential biases on CRS development remains underexplored, lacking the use of real user-involved conversation data.

### 2.2 Generative Data for Bias Mitigation

In previous literature, one commonly applied solution to address biases caused by missing data is data imputation (Hernández-Lobato et al., 2014; Steck, 2013). Data imputation estimates and generates pseudo-labels or ratings for users' unseen items, aiming to mitigate the selection bias, resulting from unfair data collection strategies. However, the performance of bias mitigation strategies based on data imputation is hindered by the issue of imputation inaccuracy, which can introduce larger biases (Wang et al., 2019). In the field of machine learning, generative neural models have advanced significantly, and the use of the generative data has become widespread for developing debiased training data (Wu et al., 2022; Lee et al., 2021).

Consequently, generative data holds the potential to mitigate biases in CRS models but has not been fully explored in previous experiments.

## 3 Bias Analysis, Formulation and Mitigation for CRS

Here we formally define the conversational recommendation task and its associated components, susceptible to various biases originating in the feedback loop. We then propose to consider the impact of multiple biases, and account for the unique characteristics of CRSs. Finally, we introduce two novel data augmentation strategies, *Once-Aug* and *PopNudge*, to mitigate the impact of biases.

### 3.1 Problem Statement

A Conversational Recommendation System (CRS) is an extension of conventional recommendation systems that enables interactive and personalised recommendations through multi-turn interactions and user feedback collection. The goal of a CRS is to effectively understand users' personal preferences and immediate intent for item exploration in order to provide natural responses that include accurate and personalised recommendations. Formally, at a specific turn $t$, for a given user $u$ among $M$ users (i.e., $U = \{u_0, u_1, ..., u_M\}$), a CRS model utilises information from the previous conversational utterances (i.e., $d = \{c_0, c_1, c_2, ..., c_{t-1}\}$), from a dialogue corpus $D$, to generate a ranked list of items $r_t$ from an item corpus $I = \{i_0, i_1, ..., i_N\}$ and formulates a natural language response $a_t$. In this study, we introduce the concept of a "conversational episode", $d^e$, inspired by reinforcement learning, which represents the conversations from when a user initiates a conversation until the recommendation is accepted by the user or the maximum interaction turns of the agent are reached (Sun and Zhang, 2018; Chu et al., 2023). The index $e$ denotes a specific episode. Consequently, we investigate the bias effects on conversational recommendations at different levels, including individual turns of recommendations ($r_t$), recommendations based on the context of a conversational episode ($r^e$), recommendations considering historical conversations $r_{t|<t}$, previous episodes $r^{e|<e}$ and corpus-level biases ($r_D$). By examining the bias effects from various perspectives, we analyse the advantages and potential impacts of the dynamic nature of CRSs.

### 3.2 Selection Bias for CRS

In the existing literature on conventional recommendation systems, selection bias is a widely examined phenomenon (e.g. Ovaisi et al., 2020; Wang et al., 2021; Liu et al., 2022). It highlights that users' behaviour in observed user-item interactions, as captured in the collected model development datasets, is often skewed, with users primarily interacting with only a small subset of items (Chen et al., 2022). Consequently, the items that are interacted within these datasets do not represent the full spectrum of available items, thereby limiting the ability of the learned model to make promising recommendations and can also exaggerate the impact of other biases, such as the popularity bias. However, the selection bias in CRSs datasets has not been thoroughly examined and discussed.

In this study, we address this gap by examining and statistically revealing the presence of selection bias in conversational recommendation datasets, focusing on their Initial Item Coverage (IIC). That is formulated as follows:

$$IIC = \frac{|I_{D_{train}}|}{|I|} \quad (1)$$

where $|\cdot|$ represents the number of unique items in the corresponding set. By calculating the IIC, we can assess the extent to which the training dataset covers the entire item space, revealing the presence and magnitude of selection bias in CRS datasets.

### 3.3 Popularity Bias

Besides selection bias, popularity bias is also a widely studied aspect when evaluating recommendation systems (Chen et al., 2023). Popularity bias arises from the tendency of users to interact more frequently with popular items, resulting in a skewed distribution where popular items receive more attention. This bias can lead to reduced coverage, diversity, and potentially lower user satisfaction levels in personalised recommendation results (Zhao et al., 2022; Abdollahpouri et al., 2017). Over time, it can also contribute to the Matthew effect, further amplifying the popularity gap between items.

A recent formalisation of popularity bias in the context of CRSs considers the recommended item list $r_{d,t}$ at turn $t$ in a dialogue (Lin et al., 2022a). The bias effect is quantified by the multiplication of two components, a ranking utility that assigns weights to items based on their ranking positions and a popularity coverage measure that indicates

the proportion of popular items included in the recommendation. The ranking utility $\pi(r_{d,t})$ is calculated as:

$$\pi(r_{d,t}) = \sum_{i \in r_u} \frac{\mathbb{1}[i \in r_u^{pop}(\eta)]}{log(rank(i)) + 1} \quad (2)$$

here, $r_u^{pop}(\eta)$ represents the set of popular items based on a popularity threshold $\eta$ determined by the items' interaction frequency. The popularity coverage $P_{r_{u,t}}$ is calculated as:

$$P(r_{u,t}) = \frac{card(r_{u,t}^{pop}(\eta))}{card(r_u)} \quad (3)$$

where $card(\cdot)$ indicates the cardinality of a set.

The above formation, as well as the existing approaches in the literature (Fu et al., 2021; Lin et al., 2022b), overlook the influence of contextual user-system interactions and the varying user intent during a conversation. Considering the multi-turn interaction nature of CRS is crucial for enhancing the user experience. In this study, we extend the formalisation of popularity bias by analysing novel factors, including cross-episode popularity and user intent-oriented popularity effects.

### 3.3.1 Cross-Episode Popularity (CEP)

Regarding the CEP effect, we assume that a rational CRS model should provide relatively independent suggestions within each individual episode of conversational recommendations. This assumption is supported by an example from an existing CRS benchmark dataset, ReDial, which is provided in the Appendix (see Figure 7). In essence, during multi-turn conversations, users often seek multiple recommendations, and we observe that the preference modelling for accurate recommendations tends to be relatively independent. Therefore, considering the dynamic preferences of users, it becomes crucial to capture their immediate interest within each episode, which may involve both popular and less popular items, to make appropriate suggestions and enhance users' satisfaction levels. To explicitly model the CEP effect, we extend the formulation of popularity bias as follows:

$$E_{CEP}[r_{d,t}^e]$$
$$= \pi(r_{d,t}^e)P_{r_{d,t}^e}|\rho(pop(r_{d,t}^e), pop(r_d^{e-1}))|$$

where $|\rho(\cdot, \cdot)|$ represents the absolute value of the Pearson correlation coefficient between the popularity levels (denoted as $pop(\cdot)$, which is a normalised score based on item frequency) of current

recommendations and the ones from the previous episodes. Diverse popularity distributions among recommended items are expected due to the different target items in each episode. A lower coefficient score indicates a weak correlation between previous episodes and current recommendations, resulting in highly diversified recommendations within a conversation interaction.

### 3.3.2 User Intent-Oriented Popularity (UIOP)

In addition to the CEP effect discussed earlier, we emphasize the importance of a CRS model in effectively capturing user intent, whether it leans towards popular items or specific preferences. By identifying and addressing user intent, CRSs can offer tailored suggestions that increase user satisfaction. To quantify the user intent-oriented popularity (UIOP) effect, we leverage the popularity of target items ($pop(i_{d,t})$) and formulate it as follows:

$$E_{UIOP}[r_{d,t}^e] = |pop(i_{d,t}) - \pi(r_{d,t}^e)P_{r_{d,t}^e}| \quad (4)$$

This equation measures the proximity between the popularity of the user's target item and the recommended items. A higher score is obtained only if the recommendations deviate from the target items.

By comprehensively evaluating multiple potential biases and effects in the development of CRSs, we enable a comprehensive assessment of conversational recommendation results. These measures promote the advancement of CRSs that prioritize fairness and unbiased recommendations, as well as boosted user experience.

### 3.4 Bias Mitigation via Generative Recommendation Dialogues

In the preceding discussion, we have identified several biases that can impact the performance of learned CRSs, which could cause an imbalanced training corpus. To mitigate the bias effects, we propose novel data augmentation approaches, joined with the generation of synthetic dialogues.

**Synthetic Data Generation.** In our generation strategy, inspired by the recent success of in-context learning on LLMs, we develop effective prompts[1] as input to the recent advanced GPT3.5 model (cf. Brown et al., 2020) to successfully generate synthetic movie recommendation dialogues. To control the frequency of item mentions, we use specific movie names as variables and ask the LMs

---

[1]Detailed prompt variants can be found in Appendix (see Figures 5 and 6).

to generate complete conversations about suggesting those movies. The generated dialogues are then reformatted to align with the input instances in the chosen benchmark.

**Data Augmentation Strategies.** We propose two data augmentation strategies to enhance model performance and mitigate multiple biases when training CRS models using the available synthetic dialogues. Previous studies, as discussed in Section 2.1, have not examined the bias effect in publicly available conversational recommendation datasets. Additionally, these studies faced challenges related to generalizability, either due to the attribute-based definition of popularity (Li et al., 2018) or the specific formulation for the Bayesian Pairwise Ranking loss (Lin et al., 2022a).

---

**Algorithm 1** PopNudge Data Augmentation

---
**Require:** $D_{train}, D_{aug}$
1: $POP_{aug} \leftarrow pop(D_{aug})$      ▷ Item popularity
2: **for** batches **do**
3:      $b_{train} \leftarrow$ batch-sample $D_{train}$
4:      $D_{aug}^{sample} \leftarrow \{\}$
5:      **for** $d_i \in b_{train}$ **do**
6:          $D_{aug}^{ret} \leftarrow D_{aug} - \{d_j, pop_j > pop_i\}$
7:          $D_{aug}^{sample} += WS(D_{aug}^{ret}, POP_{aug}, k)$
8:          ▷ $WS(C, W, k)$ is weighted sampling
9:          $k$ items from corpus $C$ with weights $W$.
10:      **end for**
11:      $D_{train} \leftarrow D_{train} \oplus D_{aug}^{sample}$
12:      model update
13: **end for**

---

In this study, we propose two data augmentation strategies. The first strategy, called 'Once-Aug' (OA), involves adding all synthetic dialogues to the training data, evenly increasing the exposure of items in the corpus. However, Once-Aug does not consider item popularity or allow control of the presence of different items. To address these issues and model a wider range of users' preferences, we introduce the 'PopNudge' strategy.

PopNudge (PN) is inspired by the data-agnostic data augmentation strategy MixUp (Zhang et al., 2018a), which augments training data with 'vicinity' instances to improve the model's ability to handle unseen inputs. Similarly, PopNudge augments training batches with dialogues recommending similar but less popular items, aiming to 'nudge' the model towards bias mitigation and avoid the frequent use of lower-rated items by assuming the

Table 1: A statistical comparison among conversational and popular non-conversational datasets. The number of dialogues and ratings are counted for conversational and non-conversational datasets, respectively. Popular items are the ratio of items with more than 5 interactions.

| Dataset | Train | Valid | Test | Items | IIC | Popular Items |
|---|---|---|---|---|---|---|
| **Conversational Recommendation Datasets** | | | | | | |
| **ReDial** | 8,103 | 900 | 1,341 | 6,112 | 72.75% | 26.55 % |
| **TGReDial** | 8,494 | 756 | 747 | 31,985 | 34.82% | 3.74% |
| **Non-Conversational Recommendation Datasets** | | | | | | |
| **ML-100k** | 100,000 | | 1,682 | 100% | 80.20% | |
| **ML-20m** | 20,000,263 | | 62,423 | 94.59% | 52.41% | |

potential relationship between popularity and ratings (Zhao et al., 2022). **Algorithm 1** outlines the procedure for applying PopNudge, augmenting the model's training batches with dialogues featuring less popular items. For every batch, we initialise an empty augmentation set (line 4). Next, for each dialogue instance, we identify its included item and prepare the set of available items with lower popularity (line 6). Then, we perform weighted sampling to select $k$ items, in addition to the augmentation set, based on item popularity (line 7). The resulting sampled items are finally augmented in the training corpus for model learning (line 11).

PopNudge offers several advantages despite its simplicity. First, it maintains the long-tail distribution of item mentions, which is consistent with natural human behaviour. This preserves the overall item distribution pattern. Secondly, it effectively increases the exposure of less popular items, leading to a more balanced item distribution. Importantly, PopNudge seamlessly integrates with existing methodologies without requiring any model updates. It selectively incorporates augmented data for model training, making it applicable in various contexts for bias mitigation purposes.

## 4 Experimental Setup

Our objective is to improve the evaluation of CRSs techniques based on various bias effects while validating the performance of our proposed data augmentation approaches. To achieve this, we conduct a series of experiments using recent CRS models on two benchmark datasets: ReDial (Li et al., 2018) and TGReDial (Zhou et al., 2020b). These experiments allow us to access the recommendation performance and bias effect on existing approaches and determine the effectiveness of our proposed approach. Table 1 presents a statistical summary of the two datasets, comparing them to the popular MovieLens (ML) datasets. The comparison reveals

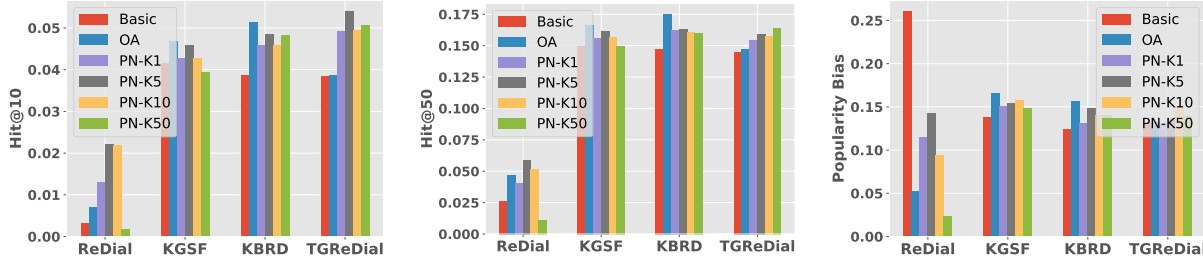

Figure 2: Performance comparison between conversational recommendation models using Hit@10 and Hit@50, together with the scores of popularity bias on the ReDial dataset. OA and PN refer to the Once Aug and PopNudge data augmentation strategies.

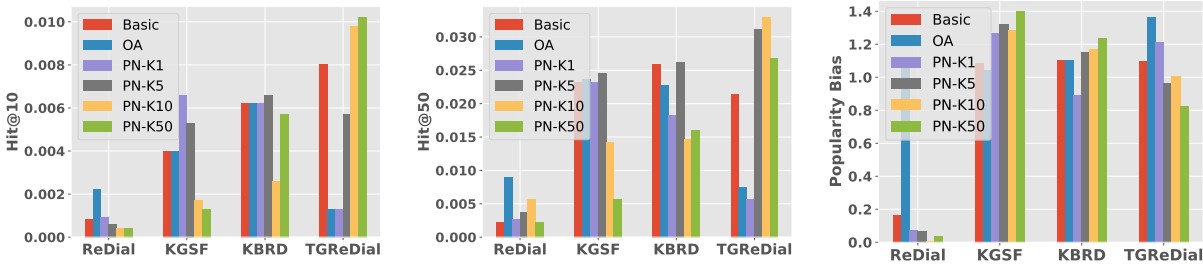

Figure 3: Performance comparison between conversational recommendation models using Hit@10 and Hit@50, together with the scores of popularity bias on the TGReDial dataset.

Table 2: Evaluation of Initial Item Coverage on two datasets and after applying data augmentation.

| Dataset | ReDial | TGReDial |
|---|---|---|
| Basic | 72.75% | 34.82% |
| OA | 100% | 100% |
| PN-$k1$ | 98.11% | 66.44% |
| PN-$k5$ | 100% | 97.81% |
| PN-$k10$ | 100% | 99.88% |
| PN-$k50$ | 100% | 100% |

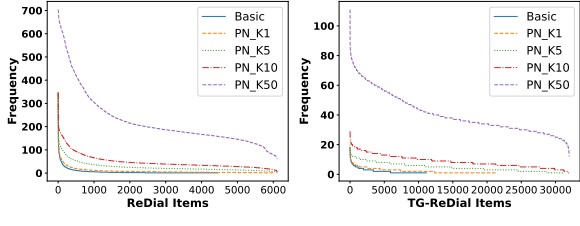

Figure 4: Mitigated Long-tail effect after applying PopNudge.

that both ReDial and TGReDial datasets are subject to selection bias, as indicated by their relatively low initial item coverage (IIC) values. For instance, only 72.75% and 34.82% of items are mentioned in the context of training dialogues in the ReDial and TGReDial datasets, respectively. Furthermore, when compared to non-conversational recommendation datasets, the benchmark conversational recommendation data contains significantly fewer interaction data. This finding further emphasizes the importance of data augmentation in addressing these limitations.

In our analysis, we consider several commonly used CRS models that serve as baselines in the literature. We summarise the recommendation component of these models: 1) **ReDial** (Li et al., 2018), an autoencoder conversational recommender and uses sentiment-analysed user opinions as input. 2) **KBRD** (Chen et al., 2019), it links entities in dialogue content to an external knowledge graph (DB-Pedia) and employs a self-attention mechanism to learn entity representations for recommendations. 3) **TG-ReDial** (Zhou et al., 2020b) uses the pretrained BERT (Kenton and Toutanova, 2019) and the sequential recommender SASRec (Kang and McAuley, 2018)) to model the historical utterances and user interactions, respectively. 4) **KGSF** (Zhou et al., 2020a) leverages both entity-oriented and word-oriented to enrich data representations and enhance item recommendations. The implementation and evaluation of these models are supported by the open-sourced CRSLab (Zhou et al., 2021) toolkit[2]. For recommendation accuracy evaluation, we follow (Zhou et al., 2020b) and apply a set of commonly used evaluation metrics, including Hit Ratio (HIT), NDCG and MRR with cutoffs at 10 and 50. To ensure fair performance comparisons, our data

---

[2]Implementation is available via: https://github.com/wangxieric/Bias-CRS

augmentation strategies solely enrich the training data and do not modify the validation and test data. Specifically, for the PopNudge strategy, we explore augmentation effects with varying numbers of sampled dialogues ($k$) ranging from {1,5,10,50}.

## 5  Result Analysis

In this section, we present and discuss the experimental results to illustrate the effectiveness of applying Once-Aug and PopNudge to baseline models while examining the recommendation accuracy, the effect on selection and popularity biases.

### 5.1  Recommendation Accuracy Evaluation

For conversational recommendation models, accurately estimating users' preferences and providing appropriate recommendations are critical. Therefore, as discussed in Section 4, in line with the existing literature, we evaluate the recommendation performance using Hit ratio, NDCG and MRR. Figures 2 and 3 depict the results of Hit@10 and Hit@50 on the ReDial and TGReDial datasets. They show that our data augmentation strategies consistently improve the Hit ratio scores, indicating enhanced recommendation accuracy. However, Once-Aug improves recommendation accuracy on the ReDial dataset but does not guarantee improvement on the TGReDial dataset. In contrast, PopNudge consistently enhances the performance of baseline CRS models on both datasets. This conclusion is also supported by the complete experimental results, joined with the evaluation using NDCG and MRR, in the Appendix (Tables 4 and 5). Thus, we conclude that PopNudge is an effective data augmentation strategy that consistently enhances recommendation accuracy with the increased exposure of available items across all four CRS models on both benchmark datasets.

### 5.2  Analysis on Selection Bias

According to the statistical summary of benchmark datasets in Table 1, CRS models are affected by selection bias in the training data. In particular, the initial CRS datasets (ReDial and TGReDial) only cover 72.75% and 34.82% of the available items, which hinders the accurate estimation of users' preferences. Therefore, we aim to comparatively evaluate the effectiveness of our data augmentation strategies in mitigating selection bias. Table 2 presents the calculated IIC scores after applying various data augmentation strategies. We

| Model | ReDial | | TGReDial | |
|---|---|---|---|---|
| | CEP | UIOP | CEP | UIOP |
| **ReDial** | 0.2000 | **0.5827** | 0.0994 | 0.6553 |
| Once-Aug | 0.0411 | 0.7693 | 0.0607 | **0.2354** |
| PopNudge_$k$1 | 0.0524 | 0.7111 | 0.0464 | 0.7482 |
| PopNudge_$k$5 | 0.1195 | 0.6858 | 0.0412 | 0.7516 |
| PopNudge_$k$10 | 0.0778 | 0.7300 | **0.0033** | 0.8123 |
| PopNudge_$k$50 | **0.0119** | 0.7976 | 0.0228 | 0.7811 |
| **KGSF** | 0.0658 | 0.6891 | 0.4301 | 0.4861 |
| Once-Aug | 0.0743 | 0.6694 | **0.3860** | **0.5069** |
| PopNudge_$k$1 | 0.0670 | 0.6848 | 0.4686 | 0.6325 |
| PopNudge_$k$5 | 0.0710 | **0.6616** | 0.4996 | 0.6574 |
| PopNudge_$k$10 | 0.0742 | 0.6733 | 0.5099 | 0.6142 |
| PopNudge_$k$50 | **0.0650** | 0.6856 | 0.6673 | 0.6123 |
| **KBRD** | **0.0574** | 0.7058 | 0.4092 | 0.4949 |
| Once-Aug | 0.0758 | **0.6750** | 0.4033 | 0.5097 |
| PopNudge_$k$1 | 0.0595 | 0.6989 | **0.3405** | **0.4558** |
| PopNudge_$k$5 | 0.0684 | 0.6832 | 0.4966 | 0.6681 |
| PopNudge_$k$10 | 0.0657 | 0.6903 | 0.6563 | 0.8557 |
| PopNudge_$k$50 | 0.0659 | 0.6900 | 0.6824 | 1.0047 |
| **TGRedial** | 0.0527 | 0.7047 | 0.4013 | 0.4565 |
| Once-Aug | 0.0529 | 0.7039 | 0.8876 | 0.5437 |
| PopNudge_$k$1 | **0.0519** | 0.7076 | 0.4626 | 0.4654 |
| PopNudge_$k$5 | 0.0554 | 0.7009 | 0.3584 | 0.4116 |
| PopNudge_$k$10 | 0.0603 | **0.6876** | 0.3724 | 0.4218 |
| PopNudge_$k$50 | 0.0564 | 0.6972 | **0.3076** | **0.3719** |

Table 3: Evaluation of Cross-Episode Popularity (CEP) and User Intent-Oriented Popularity (UIOP) effects on models and after applying data augmentation.

observe that Once-Aug easily achieves 100% coverage of all available items with equal additional exposure, while the PopNudge approach gradually increases item coverage with a growing number of sampled dialogues. Furthermore, Figure 4 demonstrates the frequency of items mentioned in the training dialogues before and after applying the PopNudge approach. It shows that the varying number of sampled dialogues ($k$) aligns with our objective discussed in Section 3.4, which aims to adjust item exposure without disrupting the long-tail distribution of natural user-item interactions. Specifically, increasing the number of sampled dialogues includes more items, with a higher frequency of popular items and gradual inclusion of less popular items. Hence, we conclude the effectiveness of both data augmentation strategies in addressing the selection bias, and PopNudge is well-performed in gradually mitigating the selection bias while retaining the long-tail distribution of user-item interactions.

### 5.3  Analysis on Popularity Bias

Next, in this study, we extensively discuss the value of investigating the popularity bias and propose additional Cross-Episode Popularity (CEP) and User Intent-Oriented Popularity (UIOP) to further examine the effectiveness of CRS models towards the

corresponding perspectives. At first, in Figure 2 and 3, we depict the scored popularity bias of several CRS models before and after applying our data augmentation strategies on ReDial and TGReDial datasets. The exact scores are also available in the Appendix (see Tables 6 and 7). According to the experimental results, we observe that a lower popularity score does not necessarily leads to a higher recommendation accuracy. For example, for the experimental results on the ReDial dataset, among the four CRS models, KGSF is the best performed with Hit@10 and Hit@50 scores at 0.0414 and 0.1496, respectively. However, it is measured with a higher popularity bias score (i.e., 0.1382) than both KBRD (0.1240) and TGReDial (0.1312). Therefore, a marginal increase in having popular items could potentially benefit the improvement of the recommendation performance of CRS models. This also has been validated in the literature (Zhao et al., 2022). In addition, as for the performance of our data augmentation strategies, we observe two main findings: (1) PopNudge has the potential to improve the model's performance when a model extremely suffers from popularity bias. This can be found in the improved recommendation accuracy but significantly lower popularity bias of the ReDial model on the ReDial dataset. (2) The increasingly sampled dialogues do not significantly increase the impact of popularity biases and can also improve the model with higher recommendation accuracy. In contrast, Once-Aug is rather unstable, such as the examined performance of the ReDial model on the TGReDial dataset.

On the other hand, we also extend the evaluation of popularity bias with two additional measures: CEP and UIOP. In Table 3, we share the full experimental results when evaluated by the CEP and UIOP scores. Accordingly, we observe that by applying our data augmentation strategies, we can lower the CEP scores in most cases apart from KBRD on the ReDial dataset. As discussed in Section 3.3.1, the CEP score can also reflect the diversity of recommendation results across conversational episodes. Hence, the observed lower CEP scores also indicate the value of our data augmentation strategies in diversifying the recommendations, which is promising in improving users' experience. Specifically, according to the experimental results in Figure 2, the initial ReDial model does not perform well on the ReDial dataset with high accuracy, which is significantly improved by our PopNudge

approach. By comparing the scores of Hit ratio and CEP, it is likely that the ReDial model suffers from a repetitive recommendation of similar items, which not only lowers the recommendation accuracy but can also harm users' experience.

At last, we examine the UIOP scores, which examine if the predictions are presenting close popularity with the target items. Similar to the findings of CEP scores, we also observe a consistent improvement towards lower UIOP scores after applying our bias mitigation strategies. In addition, we also observe that UIOP has a certain connection with recommendation accuracy. For example, KGSF boosted by PopNudge with 5 sampled dialogues is the best-performed variant as per recommendation accuracy among the basic and data-augmented models. Meanwhile, we also observe the lowest UIOP score for the identical model variant. Therefore, UIOP has the potential to serve as a metric in evaluating recommendation accuracy and joined with the consideration of item popularity.

In summary, when examined by various popularity-based measures and effects, our introduced data augmentation strategies, especially PopNudge, can effectively mitigate various biases and boost the models' performance in serving diversified and accurate recommendations. In addition, the introduction of the CEP and UIOP effects adds additional insights to the performance evaluation of CRS techniques and can also be leveraged in the future development of CRS models.

## 6 Conclusions

Our study investigates several biases within the feedback loop of developing a conversational recommendation system, including selection bias, popularity bias, cross-episode popularity (CEP) effect and user intent-oriented popularity (UIOP) effect. These proposed biases and effects shed light on CRS model performance and provide valuable insights for effective model development. Additionally, we leverage the recent advance of large language models to generate effective synthetic data and propose novel and effective data augmentation strategies, Once-Aug and PopNudge, to enhance model performance. Our experimental results demonstrate the effectiveness of our data augmentation strategies in improving CRS model performance while considering the recommendation accuracy as well as the mitigation across selection bias and popularity biases.

## Limitations

In this paper, we make significant contributions by utilising generative dialogues to augment the training data and mitigate biases. However, we acknowledge two main limitations of our approach. Firstly, our generative dialogues only involve a single item per dialogue, which deviates from the majority of dialogues in the initial training corpus and may impact cross-episode recommendations. Secondly, we primarily focus on improving recommendation performance and overlook the evaluation of responses, an essential aspect of conversational techniques. Addressing these limitations will require further dedicated efforts in future work.

## Acknowledgements

This research is supported by the Alan Turing Institute under the EPSRC grant [EP/N510129/1], the EPSRC Fellowship titled "Task Based Information Retrieval" [EP/P024289/1] and the Engineering and Physical Sciences Research Council [EP/S021566/1]. Jiqun Liu's participation in this project is partially supported by the National Science Foundation (NSF) Award IIS-2106152 and a Junior Faculty Fellowship Program Award from the University of Oklahoma Office of the Vice President for Research and Partnerships.

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

# A   In-Context Learning Prompts for Dataset Generation

You help in generating a mix-initiative synthetic multi-turn conversational recommendation dialog, each utterance starts with [user] or [system], by leveraging a mentioned movie as input.

**Example movie**: [The Shawshank Redemption]
**Dialogue**:

[user]: I'm looking for a good movie. Any suggestions?

[system]: Absolutely! I highly recommend The Shawshank Redemption. It's a critically acclaimed film known for its powerful storytelling and exceptional performances.

[user]: Oh, I think I've heard of it. What's it about?

[system]: The Shawshank Redemption tells the story of Andy Dufresne, a banker wrongly convicted of murder and his experiences in Shawshank State Penitentiary. It's a compelling tale of hope, friendship, and the strength of the human spirit.

[user]: That sounds intriguing. Is it a drama or more of a thriller?

[system]: It's primarily a drama, but it has elements of both genres. The film balances intense moments with emotional depth, creating a captivating and thought-provoking experience.

[user]: Great! I'll give it a watch then. Thanks for the recommendation!

[system]: You're welcome! I'm sure you'll enjoy "The Shawshank Redemption." If you have any more questions or need further recommendations, feel free to ask. Enjoy the movie!

**Input**: [movie name]

Figure 5: In-Context Learning Prompts for generating synthetic recommendation dialogue for augmenting ReDial dataset.

您帮助自动生成一个多轮次的电影推荐对话，每个发言以[用户]或[系统]开头，并利用提供的电影名作为输入。

**样例:** 电影名 [十面埋伏]
**生成对话:**

[用户]: 嗨，我正在寻找一部好电影看。你有什么推荐吗？

[系统]: 当然！你有任何特定类型或题材的电影偏好吗？

[用户]: 我想看一部动作武侠电影。有什么好的选择吗？

[系统]: 嗯，如果你对动作武侠感兴趣，我可以向你推荐一部叫做《十面埋伏》的电影。这部经典电影充满了精彩的武打动作和紧张的情节。

[用户]: 听起来不错！请告诉我更多关于《十面埋伏》的信息。

[系统]: 《十面埋伏》是由中国导演张艺谋执导的一部史诗动作片。影片讲述的是晚唐时期两个捕快与一个歌妓的爱情故事。原本歌妓喜欢的是刘捕头，但后来经过一番周折，歌妓发现自己最爱的竟是金捕头，刘捕头则因爱生恨，萌发了杀死歌妓的念头。最后歌妓为了保住金捕头的性命，与刘捕头同归于尽。

[用户]：太棒了！这个推荐都很适合我。谢谢你的帮助！

[系统]：不客气！我很高兴能帮助你找到适合你口味的电影。希望你享受这些推荐，如果你还有其他问题，随时告诉我！

**电影名:** [电影]

Figure 6: In-Context Learning Prompts for generating synthetic recommendation dialogue for augmenting TG-ReDial dataset. TG-ReDial includes dialogues in the language of Chinese, which leads to the use of a Chinese prompt.

## B  Dialogue Example for Motivation

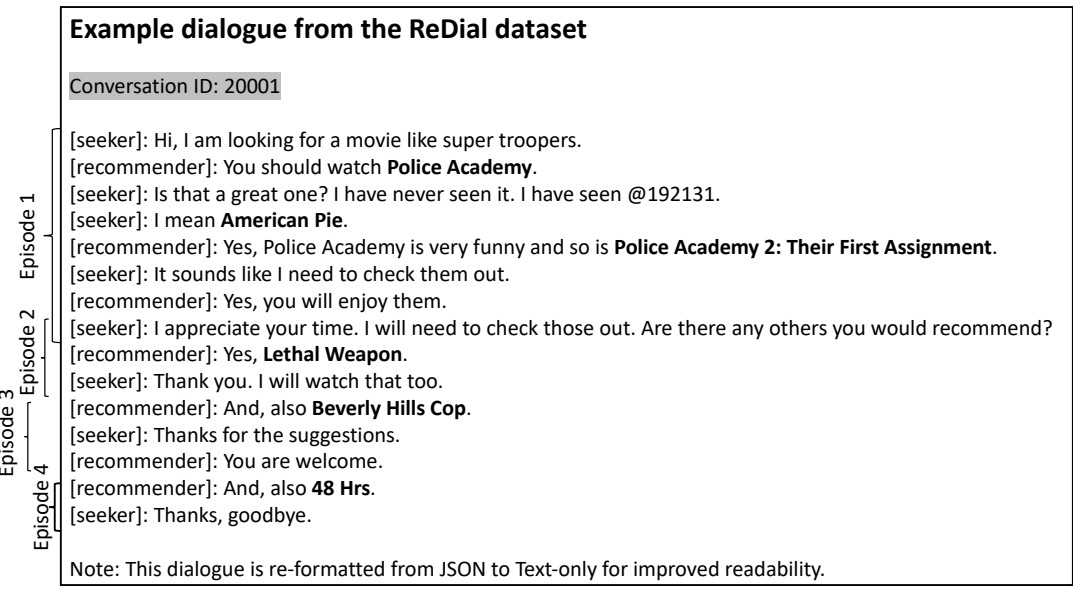

**Example dialogue from the ReDial dataset**

Conversation ID: 20001

[seeker]: Hi, I am looking for a movie like super troopers.
[recommender]: You should watch **Police Academy**.
[seeker]: Is that a great one? I have never seen it. I have seen @192131.
[seeker]: I mean **American Pie**.
[recommender]: Yes, Police Academy is very funny and so is **Police Academy 2: Their First Assignment**.
[seeker]: It sounds like I need to check them out.
[recommender]: Yes, you will enjoy them.
[seeker]: I appreciate your time. I will need to check those out. Are there any others you would recommend?
[recommender]: Yes, **Lethal Weapon**.
[seeker]: Thank you. I will watch that too.
[recommender]: And, also **Beverly Hills Cop**.
[seeker]: Thanks for the suggestions.
[recommender]: You are welcome.
[recommender]: And, also **48 Hrs**.
[seeker]: Thanks, goodbye.

Note: This dialogue is re-formatted from JSON to Text-only for improved readability.

Figure 7: Dialogue example from the ReDial dataset, which shows the rather independent episode-wise interaction between user and system, for the motivation of investigating cross-episode popularity effect.

# C  Collection of Full Experimental Results

Table 4: Initial Experimental results of conversational recommenders on the **ReDial** dataset.

| Model | Hit@10 | Hit@50 | NDCG@10 | NDCG@50 | MRR@10 | MRR@50 |
|---|---|---|---|---|---|---|
| **ReDial** | 0.0032 | 0.0262 | 0.0010 | 0.0061 | 0.0004 | 0.0016 |
| - Once-Aug | 0.0069 | 0.0464 | 0.0042 | 0.0141 | 0.0033 | **0.0061** |
| - PopNudge_$k$1 | 0.0129 | 0.0404 | 0.0067 | 0.0126 | 0.0048 | 0.0059 |
| - PopNudge_$k$5 | 0.0220 | 0.0584 | 0.0103 | 0.0179 | 0.0067 | 0.0081 |
| - PopNudge_$k$10 | **0.0219** | **0.0517** | **0.0107** | **0.0172** | **0.0072** | **0.0085** |
| - PopNudge_$k$50 | 0.0017 | 0.0109 | 0.0006 | 0.0026 | 0.0002 | 0.0007 |
| **KGSF** | 0.0414 | 0.1496 | 0.0229 | 0.0476 | 0.0168 | 0.0227 |
| - Once-Aug | **0.0467** | **0.1658** | **0.0257** | **0.0529** | 0.0189 | 0.0252 |
| - PopNudge_$k$1 | 0.0427 | 0.1554 | 0.0238 | 0.0496 | 0.0176 | 0.0237 |
| - PopNudge_$k$5 | 0.0457 | 0.1614 | **0.0257** | 0.0522 | **0.0191** | **0.0254** |
| - PopNudge_$k$10 | 0.0427 | 0.1569 | 0.0235 | 0.0493 | 0.0173 | 0.0232 |
| - PopNudge_$k$50 | 0.0394 | 0.1496 | 0.0218 | 0.0470 | 0.0161 | 0.0220 |
| **KBRD** | 0.0387 | 0.1471 | 0.0209 | 0.0459 | 0.0151 | 0.0211 |
| - Once-Aug | **0.0514** | **0.1749** | **0.0287** | **0.0569** | **0.0212** | **0.0278** |
| - PopNudge_$k$1 | 0.0457 | 0.1621 | 0.0254 | 0.0518 | 0.0187 | 0.0249 |
| - PopNudge_$k$5 | 0.0484 | 0.1631 | 0.0272 | 0.0536 | 0.0202 | 0.0265 |
| - PopNudge_$k$10 | 0.0459 | 0.1609 | 0.0257 | 0.0518 | 0.0190 | 0.0251 |
| - PopNudge_$k$50 | 0.0483 | 0.1599 | 0.0272 | 0.0526 | 0.0202 | 0.0262 |
| **TGReDial** | 0.0384 | 0.1442 | 0.0214 | 0.0458 | 0.0158 | 0.0217 |
| - Once-Aug | 0.0387 | 0.1466 | 0.0216 | 0.0464 | 0.0159 | 0.0219 |
| - PopNudge_$k$1 | 0.0492 | 0.1541 | 0.0271 | 0.0512 | 0.0199 | 0.0257 |
| - PopNudge_$k$5 | **0.0539** | **0.1590** | **0.0306** | **0.0550** | **0.0228** | **0.0289** |
| - PopNudge_$k$10 | 0.0494 | 0.1574 | 0.0272 | 0.0523 | 0.0200 | 0.0261 |
| - PopNudge_$k$50 | 0.0507 | 0.1638 | 0.0289 | 0.0554 | 0.0217 | 0.0283 |

Table 5: Experimental results of conversational recommenders on the **TGReDial** dataset.

| Model | Hit@10 | Hit@50 | NDCG@10 | NDCG@50 | MRR@10 | MRR@50 |
|---|---|---|---|---|---|---|
| **ReDial** | 0.0008 | 0.0022 | 0.0004 | 0.0007 | 0.0002 | 0.0003 |
| - Once-Aug | **0.0022** | **0.0089** | **0.0009** | **0.0025** | **0.0006** | **0.0009** |
| - PopNudge_$k$1 | 0.0009 | 0.0027 | 0.0005 | 0.0009 | 0.0004 | 0.0005 |
| - PopNudge_$k$5 | 0.0006 | 0.0037 | 0.0002 | 0.0009 | 0.0001 | 0.0003 |
| - PopNudge_$k$10 | 0.0004 | 0.0057 | 0.0002 | 0.0013 | 0.0001 | 0.0003 |
| - PopNudge_$k$50 | 0.0004 | 0.0022 | 0.0002 | 0.0006 | 0.0001 | 0.0002 |
| **KGSF** | 0.0040 | 0.0231 | 0.0022 | 0.0063 | 0.0016 | 0.0025 |
| - Once-Aug | 0.0040 | 0.0236 | 0.0023 | 0.0068 | 0.0018 | 0.0028 |
| - PopNudge_$k$1 | **0.0066** | 0.0231 | **0.0034** | 0.0073 | **0.0024** | 0.0033 |
| - PopNudge_$k$5 | 0.0053 | **0.0245** | 0.0031 | **0.0075** | **0.0024** | **0.0034** |
| - PopNudge_$k$10 | 0.0017 | 0.0142 | 0.0008 | 0.0037 | 0.0005 | 0.0012 |
| - PopNudge_$k$50 | 0.0013 | 0.0057 | 0.0007 | 0.0016 | 0.0005 | 0.0006 |
| **KBRD** | 0.0062 | 0.0258 | 0.0033 | 0.0077 | 0.0024 | 0.0034 |
| - Once-Aug | 0.0062 | 0.0227 | 0.0036 | 0.0076 | 0.0028 | 0.0038 |
| - PopNudge_$k$1 | 0.0062 | 0.0182 | 0.0037 | 0.0067 | 0.0028 | 0.0037 |
| - PopNudge_$k$5 | **0.0066** | **0.0261** | **0.0040** | **0.0081** | **0.0032** | **0.0041** |
| - PopNudge_$k$10 | 0.0026 | 0.0147 | 0.0014 | 0.0046 | 0.0011 | 0.0019 |
| - PopNudge_$k$50 | 0.0057 | 0.0160 | 0.0031 | 0.0054 | 0.0023 | 0.0027 |
| **TGReDial** | 0.0080 | 0.0213 | 0.0045 | 0.0076 | 0.0033 | 0.0041 |
| - Once-Aug | 0.0013 | 0.0075 | 0.0007 | 0.0021 | 0.0005 | 0.0008 |
| - PopNudge_$k$1 | 0.0013 | 0.0057 | 0.0006 | 0.0057 | 0.0004 | 0.0007 |
| - PopNudge_$k$5 | 0.0057 | 0.0311 | 0.0033 | 0.0091 | 0.0025 | 0.0039 |
| - PopNudge_$k$10 | 0.0098 | **0.0329** | 0.0056 | **0.0108** | 0.0042 | **0.0054** |
| - PopNudge_$k$50 | **0.0102** | 0.0267 | **0.0059** | 0.0096 | **0.0045** | 0.0053 |

Table 6: Popularity bias analysis of Conversational Recommendations on the **ReDial** Dataset

| Model | Pop Bias (Mean) | Pop Bias (Std) |
|---|---|---|
| **ReDial** | 0.2601 | 0.0005 |
| - Once-Aug | 0.0523 | 0.0001 |
| - PopNudge_$k$1 | 0.1149 | 0.0364 |
| - PopNudge_$k$5 | 0.1419 | 0.0001 |
| - PopNudge_$k$10 | 0.0940 | 0.0045 |
| - PopNudge_$k$50 | 0.0229 | 0.0005 |
| **KGSF** | 0.1382 | 0.1183 |
| - Once-Aug | 0.1654 | 0.1501 |
| - PopNudge_$k$1 | 0.1499 | 0.1364 |
| - PopNudge_$k$5 | 0.1540 | 0.1463 |
| - PopNudge_$k$10 | 0.1576 | 0.1471 |
| - PopNudge_$k$50 | 0.1483 | 0.1513 |
| **KBRD** | 0.1240 | 0.1208 |
| - Once-Aug | 0.1565 | 0.1142 |
| - PopNudge_$k$1 | 0.1308 | 0.1257 |
| - PopNudge_$k$5 | 0.1484 | 0.1279 |
| - PopNudge_$k$10 | 0.1403 | 0.1195 |
| - PopNudge_$k$50 | 0.1398 | 0.1128 |
| **TGReDial** | 0.1312 | 0.1155 |
| - Once-Aug | 0.1320 | 0.1064 |
| - PopNudge_$k$1 | 0.1292 | 0.1074 |
| - PopNudge_$k$5 | 0.1367 | 0.1099 |
| - PopNudge_$k$10 | 0.1509 | 0.1259 |
| - PopNudge_$k$50 | 0.1388 | 0.1178 |

Table 7: Popularity bias analysis of Conversational Recommendations on the **TGReDial** Dataset

| Model | Pop Bias (Mean) | Pop Bias (Std) |
|---|---|---|
| **ReDial** | 0.1626 | 0.0013 |
| - Once-Aug | 1.0534 | 0.0307 |
| - PopNudge_$k$1 | 0.0697 | 0.0001 |
| - PopNudge_$k$5 | 0.0663 | 0.0321 |
| - PopNudge_$k$10 | 0.0056 | 0.0001 |
| - PopNudge_$k$50 | 0.0368 | 0.0104 |
| **KGSF** | 1.0847 | 0.5789 |
| - Once-Aug | 1.0438 | 0.6240 |
| - PopNudge_$k$1 | 1.2668 | 0.6837 |
| - PopNudge_$k$5 | 1.3174 | 0.6899 |
| - PopNudge_$k$10 | 1.2844 | 0.6122 |
| - PopNudge_$k$50 | 1.3958 | 0.5049 |
| **KBRD** | 1.1012 | 0.5925 |
| - Once-Aug | 1.1005 | 0.6144 |
| - PopNudge_$k$1 | 0.8892 | 0.5747 |
| - PopNudge_$k$5 | 1.1514 | 0.6588 |
| - PopNudge_$k$10 | 1.1700 | 0.7493 |
| - PopNudge_$k$50 | 1.2343 | 0.7804 |
| **TGReDial** | 1.0943 | 0.5372 |
| - Once-Aug | 1.3614 | 0.1649 |
| - PopNudge_$k$1 | 1.2120 | 0.3565 |
| - PopNudge_$k$5 | 0.9601 | 0.5202 |
| - PopNudge_$k$10 | 1.0031 | 0.5239 |
| - PopNudge_$k$50 | 0.8259 | 0.4630 |