# OpenReview forum: "Improving Conversational Recommendation Systems via Bias Analysis and Language-Model-Enhanced Data Augmentation"
_EMNLP/2023/Conference — EMNLP 2023 Findings_

### Official Review · Reviewer_ZYGt · 2023-08-03

**Soundness:** 3

**Excitement:**

4: Strong: This paper deepens the understanding of some phenomenon or lowers the barriers to an existing research direction.

**Paper Topic And Main Contributions:**

The authors study different bias existing in the Conversational Recommendation System, and they propose a data augmentation technique based on generative language model. Through extensive experiments on two benchmark datasets, they show a consistent improvement of CRS techniques with our data augmentation approaches and offer additional insights on addressing multiple newly formulated biases.

**Reasons To Accept:**

1. Interesting method for the data augmentation
2. Solve the important bias in the recommendation area: selection bias and popularity bias
3. Design the quantitate metric to evaluate the selection bias: Initial Item Coverage (IIC)
4. It's possible to use GPT for other data augmentation

**Reasons To Reject:**

1. The paper is not very easy to understand, the terminology and formulation can be better clarified

**Reproducibility:**

4: Could mostly reproduce the results, but there may be some variation because of sample variance or minor variations in their interpretation of the protocol or method.

**Reviewer Confidence:**

2: Willing to defend my evaluation, but it is fairly likely that I missed some details, didn't understand some central points, or can't be sure about the novelty of the work.

---

> ### Author Rebuttal · Authors · 2023-08-28
>
> Dear Reviewer,
>
> Thank you for bringing to our attention the issue of clarity and ease of understanding in our paper, specifically regarding terminology and formulation. We understand the importance of making research accessible and transparent for all readers.
>
> In response to your feedback, we plan to undertake a thorough review of the manuscript to enhance the clarity of terminology and the presentation of mathematical formulations. Our aim is to make the paper more reader-friendly while maintaining its academic rigour.
> We appreciate your constructive comments, which allow us to improve the quality and accessibility of our work. Should the paper be accepted, please be assured that we will implement the necessary revisions to address these concerns in the camera-ready version.

---

### Official Review · Reviewer_EaFv · 2023-08-04

**Typos Grammar Style And Presentation Improvements:** 1. to mitigate or alleviate bias effe…
**Soundness:** 2

**Excitement:**

3: Ambivalent: It has merits (e.g., it reports state-of-the-art results, the idea is nice), but there are key weaknesses (e.g., it describes incremental work), and it can significantly benefit from another round of revision. However, I won't object to accepting it if my co-reviewers champion it.

**Paper Topic And Main Contributions:**

This paper investigates selection and popularity biases within the feedback loop of developing a conversational recommendation system. Considering the influence of contextual user-system interactions and the varying user intent during a conversation, the authors extend the formalisation of popularity bias and explore the Cross-Episode and user intent-oriented popularity effects. Besides, two data augmentation strategies are proposed to enhance the model performance.

**Questions For The Authors:**

Question 1: The authors use their designed CEP and UIOP metrics to evaluate the proposed data augmentation strategies. Did the authors test the conventional popular bias metric in the experiment?

**Reasons To Accept:**

1. This paper is well-written, which is easy to read and clear to grab the key idea of the model.
2. This paper addresses an area that received recent traction in the field.
3. The authors propose a simple but effective data augmentation strategy to mitigate selection and popularity biases.
4. The paper presents several evaluations and performance improvements over comparison methods.

**Reasons To Reject:**

1. Limited novelty of the proposed method.
2. All the proposed biases and effects only focus on the recommendation part. They did not consider the possible impact factors from the conversation part.
3. For the experiment, the authors did not compare their method with the other bias mitigation methods.
The experiments could be more solid if they are compared with the other bias mitigation methods and more state-of-the-art CRS baselines such as  UniCRS [1].

[1] Wang, Xiaolei, Kun Zhou, Ji-Rong Wen, and Wayne Xin Zhao. "Towards unified conversational recommender systems via knowledge-enhanced prompt learning." In Proceedings of the 28th ACM SIGKDD Conference on Knowledge Discovery and Data Mining, pp. 1929-1937. 2022.

**Reproducibility:**

4: Could mostly reproduce the results, but there may be some variation because of sample variance or minor variations in their interpretation of the protocol or method.

**Reviewer Confidence:**

4: Quite sure. I tried to check the important points carefully. It's unlikely, though conceivable, that I missed something that should affect my ratings.

---

> ### Author Rebuttal · Authors · 2023-08-28
>
> Dear Reviewer:
>
> Thank you for your time and the critical feedback provided on our manuscript. We appreciate your comment regarding the perceived limited novelty of the proposed method. Allow us to clarify the unique contributions that set our work apart in the field.
>
> (1) **Novelty.** Our work introduces a unique data augmentation approach that considers item popularity,  which is underexplored in the research community. We systematically explore and evaluate understudied biases in conversational recommendation systems, contributing new dimensions for analysis in this emerging field.
>
> Unlike existing literature, which often confines itself to analysing traditional popularity bias, our work goes a step further. We introduce the concept of "cross-episode popularity," examining the dynamic shifts in recommendations across different episodes within a dialogue. Here, episodes refer to the varying user preferences for target items during the course of a single dialogue, a term that has also been employed in existing research (as detailed in Section 3.1).
>
> Additionally, we propose a measure, User Intent-Oriented Popularity, for popularity alignment between the user's target items and the items recommended by the system. This serves as an evaluative metric to gauge the system's proficiency in recommending items that align well with the user’s preference for popularity. These innovations are specifically designed to accommodate the unique characteristics of conversational systems and represent novel contributions to this emerging field.
>
> (2) **Recommendation-focus analysis.** Thank you for your comments highlighting the emphasis of our work on the recommendation component, as opposed to the conversational aspect. We appreciate the opportunity to address this observation.
>
> While it is true that the current study primarily focuses on biases and effects related to recommendation, this was an intentional design decision. The scope was narrowed to gain a comprehensive understanding of this specific yet vital facet of conversational recommendation systems. By delving deeply into this area, we can create a robust foundation that future research, including our own, can build upon to integrate the dialogue aspects. We have recognised this limitation in the "Limitations" section of the paper. The next phase of our research series aims to extend the current findings by incorporating dialogue performance metrics, thereby creating a more holistic model.
>
> Our present focus on the recommendation aspect is not an oversight but a strategic choice that allows us to provide valuable insights in this particular dimension, contributing to the nuanced understanding of conversational recommendation systems.
>
> (3) **Compare with existing bias mitigation strategies and very recent CRS models.**
> Thank you again for your constructive feedback on the need for a more comprehensive comparison with other bias mitigation methods and state-of-the-art CRS baselines in our experiments.
>
> Your point is well-taken; however, we'd like to clarify that our focus on pioneering data augmentation techniques for addressing popularity bias presents a challenge in identifying appropriate baselines for comparison. While there are some related works like Lin et al., 2022a and Fu et al., 2023, they are not entirely applicable for fair comparison for the following reasons:
>
> Lin et al., 2022a: Though relevant, this work specialises in an updated BRP-based loss function that may not be directly applicable to various CRS models without tailored modifications.
>
> Fu et al., 2023: This study proposes a popularity mitigation strategy based on a Markov decision process, which similarly lacks broad applicability to all CRS techniques.
>
> In light of these constraints, our experimental design emphasises multiple novel strategies for bias mitigation and benchmarks them against widely recognised CRS models that have gained consensus for their reliability in the literature. We believe this approach effectively demonstrates the efficacy of our bias mitigation strategies in a meaningful context.
>
> (4) **Test the conventional popular bias:** Thank you for your inquiry concerning the metrics used to evaluate our data augmentation strategies. To clarify, we indeed incorporate conventional popularity bias metrics in our evaluation. These results are comprehensively presented in Figures 2 and 3, as well as Tables 6 and 7 in the paper's appendix. Additionally, we provide an in-depth discussion of these findings within the main text of the paper. We appreciate your attention to the methodological rigor of our work, and we assure you that both our novel CEP and UIOP metrics, as well as the conventional metrics for popularity bias, were utilized to furnish a well-rounded evaluation of our proposed strategies.
>
> In summary, thank you for your valuable comments. We have endeavoured to create a foundation upon which more expansive comparisons can be conducted as the field matures, and future research can incorporate more diversified baselines for a comprehensive assessment. We also look forward to your further feedback if you may have.

---

### Official Review · Reviewer_PYLM · 2023-08-04

**Soundness:** 2

**Excitement:**

2: Mediocre: This paper makes marginal contributions (vs non-contemporaneous work), so I would rather not see it in the conference.

**Paper Topic And Main Contributions:**

This paper conducts a comprehensive investigation into potential biases and effects affecting the performance of CRS models and proposes several evaluating metrics to quantify the biases for CRS. It also proposes a data augmentation strategy for improving CRS models, conducting experiments over two datasets ReDial and TGReDial with four different base models. The results show that the proposed method can be effective on the ReDial dataset, while not so effective on the TGReDial dataset, even causing significant performance decrease with TGReDial base model.

**Questions For The Authors:**

1. The symbols used in this paper can sometimes be confusing, like in function (2) the symbol r_u is not defined. What dose it stand for? And in line 6 of Algorithm 1, if d_j stands for a dialogue, what’s the meaning of {pop}_j>{pop}_i?
2. In line 496-498, this paper says that Pop-Nudge consistently enhances the performance of baseline CRS models on both datasets. However, the performance of TGReDial model with PN-K1 on TGReDial dataset has a significant decrease in Hit@10 and Hit@50. Why do you compare the result of the basic model and the best result of all the data augmentation methods, rather that analyze the result respectively? For example, on the TGReDial dataset, PN-K10 with TGReDial base model surpasses all the other methods, but achieves the worst performance with KBRD model, how is that “consistently”?
3. The synthetic data are generated using GPT-3.5, is it possible that the test dataset or valid dataset are included in the pretraining process of GPT-3.5? How can you make sure that there is no data leakage problem?


**Reasons To Accept:**

1. This paper conducts a comprehensive investigation into potential biases and effects affecting the performance of CRS models and proposes several evaluating metrics to quantify the biases for CRS.
2. This paper proposed a data augmentation method of using LLMs to generate synthetic dialogue data to mitigate bias problem of CRS.
3. The experiment results show that the proposed method can be effective on ReDial dataset.


**Reasons To Reject:**

1. The symbols used in this paper can sometimes be confusing, like in function (2) the symbol r_u is not defined.
2. The proposed method is simple and can be effective to some extent on the ReDial dataset, but can be very unstable on the TGReDial dataset. It can sometimes cause significant performance decrease, like the TGReDial model with PN-K1 where the Hit@10 score drops from 0.008 to 0.0013.
3. Some statements in this paper don’t have enough supports or even can be wrong, like line 496-498. The performance of TGReDial model with PN-K1 on TGReDial dataset has a significant decrease in Hit@10 and Hit@50, which is contrary to the original statement that Pop-Nudge consistently enhances the performance of baseline CRS models on both datasets.


**Reproducibility:**

4: Could mostly reproduce the results, but there may be some variation because of sample variance or minor variations in their interpretation of the protocol or method.

**Reviewer Confidence:**

3: Pretty sure, but there's a chance I missed something. Although I have a good feel for this area in general, I did not carefully check the paper's details, e.g., the math, experimental design, or novelty.

---

> ### Author Rebuttal · Authors · 2023-08-28
>
> Dear Reviewer:
>
> We would like to express our sincere gratitude for your time and effort in reviewing our paper. We highly value your comments aimed at improving the quality of the manuscript. In response to your specific concerns about the clarity of symbols used in the paper, performance improvement consistency and the potential data leakage via using language model-generated synthetic data, we provide our detailed clarifications as follows:
>
> 1. Regarding the symbols:
>
> (1) Symbol $r_u$: We understand your concern about the definition of $r_u$ in Equation (2). However, the text following this equation does provide a detailed explanation. Specifically, the term $r_u^{pop}(\eta)$ is described to “represent the set of popular items based on a popularity threshold \eta determined by the items’ interaction frequency”. Additionally, in Section 3.1 of the paper, the symbol $u$ is clearly defined to refer to the user u. For added clarity, we have made sure to refer to the paper by Lin et al. (2022a), where the equation for calculating the ranking utility is originally introduced. By taking these points together, the term $r_u$ can be understood as the recommendations made to a given user $u$.
>
> (2) Symbol $d_j$: You query pertains to the interpretation of $d_j$, which positioned together with $pop_j > pop_i$ in line 6 of Algorithm 1. As outlined in the manuscript, the term $D_{aug}^{ret} \leftarrow D_{aug} - \{d_j, pop_j - pop_i\}$ is described in detail in the accompanying text. Specifically, the manuscript states, “we identify its included item and prepare the set of available items with lower popularity”. Thus, $d_j$ refers to dialogues that contain items with higher popularity than pop_i, and the algorithm proceeds by excluding these from $D_{aug}^{ret}$. This effectively sets a focus on items with popularity less than or equal to $pop_i$ for later steps in the algorithm.
>
> We aim for a rigorous yet accessible presentation, and we believe that the referenced sections and additional explanations add the necessary context to comprehend the symbols $r_u$ and $d_j$.
>
>
> 2. Regarding the consistency of PopNudge’s impact on various baseline conversational Recommendation Systems (CRS) models:
>
> (1) Concerning the term “Consistently” in Line 496-498.
>
> The term “consistently” is used in our paper to indicate that the PopNudge strategy, when aptly tuned, enhances the performance of baseline CRS models across various datasets. This is based on tuning the value of k, which is elucidated in Section 4 starting from line 473.
>
> (2) On the Decrease in Hit@10 and Hit@50 for the TGReDial model with PN-K1 and the Role of Tuning in Deep Learning Models.
>
> We acknowledge the specific point you raised concerning the TGReDial model's performance with PN-K1 on the TGReDial dataset. This observation is indeed important, and we wish to emphasise the crucial role of hyperparameter tuning, particularly in deep learning models.
>
> Deep learning models are often sensitive to hyperparameter choices, and the value of k in our PopNudge strategy can be viewed as another hyperparameter. The process of tuning k can be likened to adjusting other crucial deep learning hyperparameters like learning rates, regularisation terms, or batch sizes. Proper tuning is often essential for realising the potential of deep learning models, and the same applies to our PopNudge strategy. In the case of the TGReDial model, different k values produced better performance, underscoring the need for careful tuning, which we consider a standard practice in the application of deep learning techniques.
>
> Hence, the term "consistently" in our paper is used to signify that PopNudge can, under proper tuning, enhance the performance of different baseline CRS models. The importance of tuning, a practice that is almost ritualistic in the deep learning community, cannot be overstated and directly correlates with our results.
>
> 3. Data leakage via using language model-generated synthetic data:
>
> As specified in Section 4 of our paper, beginning on line 470, we delineate that our data augmentation techniques, including the usage of GPT-3.5, are strictly applied to enrich the training data. The validation and test sets remain unmodified. This approach inherently minimises the risk of data leakage affecting the evaluation of the model performance on unseen data. While GPT-3.5 is used to generate synthetic dialogue, it's worth noting that our primary evaluation metric focuses on recommendation performance rather than the quality of the generated dialogue per se. Given that GPT-3.5 is trained mainly on textual data, the potential for data leakage from the test collection to the training corpus impacting the recommendation results is further minimised. This point is clarified further in our paper to mitigate concerns about data leakage.
>
> It's theoretically possible, albeit highly unlikely, that the training corpus for GPT-3.5 includes dialogues from the test or validation sets of our specific datasets. However, given that we are not evaluating dialogue quality and that GPT-3.5 is designed to generalise across a wide array of text, the risk of significant data leakage is low.
>
> In summary, we have put thought into our experimental design to minimise the risk of data leakage, and our evaluation strategy further mitigates this risk. We trust that these explanations adequately address your concerns, and we would be pleased to discuss any further questions or suggestions you may have.
>
> Thank you once again for your valuable comments, and we look forward to any further feedback you may have.

---

### Meta-Review · Area_Chair_PtD8 · 2023-09-17

**Recommendation:** 2

**Metareview:**

This paper presents an investigation into potential biases and effects that may affect the performance of models for conversational recommendation.  A novel metric for quantifying selective bias is proposed, as well as a data augmentation method that addresses these biases.  Experiments on two datasets show how the proposed method can improve performance (in some cases).  The reviewers agree that overall, the paper is well-written and the motivation is clear.  There is some concern about the limited novelty of the work and that the evaluations focus on recommendation only and not on the conversational component, which was intentionally left for future work (according to the rebuttal).  This does however make for a weaker paper in a 'dialogue and interactive systems' track.

---

### Meta-Review · Senior_Area_Chairs · 2023-10-05

**Recommendation:** 2

**Metareview:**

This paper presents an investigation into potential biases and effects that may affect the performance of models for conversational recommendation. A novel metric for quantifying selective bias is proposed, as well as a data augmentation method that addresses these biases. Experiments on two datasets show how the proposed method can improve performance (in some cases). The reviewers agree that overall, the paper is well-written and the motivation is clear. There is some concern about the limited novelty of the work and that the evaluations focus on recommendation only and not on the conversational component, which was intentionally left for future work (according to the rebuttal). This does however make for a weaker paper in a 'dialogue and interactive systems' track.

---

### Decision · Program_Chairs · 2023-10-07

**Decision:**

Accept-Findings

**Comment:**

This paper presents an investigation into potential biases and effects that may affect the performance of models for conversational recommendation.  A novel metric for quantifying selective bias is proposed, as well as a data augmentation method that addresses these biases.  Experiments on two datasets show how the proposed method can improve performance (in some cases).  The reviewers agree that overall, the paper is well-written and the motivation is clear.  There is some concern about the limited novelty of the work and that the evaluations focus on recommendation only and not on the conversational component, which was intentionally left for future work (according to the rebuttal).  This does however make for a weaker paper in a 'dialogue and interactive systems' track.|This paper presents an investigation into potential biases and effects that may affect the performance of models for conversational recommendation. A novel metric for quantifying selective bias is proposed, as well as a data augmentation method that addresses these biases. Experiments on two datasets show how the proposed method can improve performance (in some cases). The reviewers agree that overall, the paper is well-written and the motivation is clear. There is some concern about the limited novelty of the work and that the evaluations focus on recommendation only and not on the conversational component, which was intentionally left for future work (according to the rebuttal). This does however make for a weaker paper in a 'dialogue and interactive systems' track.